# Holocene hydrological changes of the Rhone River (NW Mediterranean) as recorded in the marine mud belt

**Bassetti, M.A. (1), Berné S. (1), Sicre M.-A. (2), Dennielou B.(3), Alonso. Y. (1), Buscail R. (1), Jalali, B. (4) ; Hebert B. (1), C. Menniti (1)**

(1) CEFREM UMR5110 CNRS, Université de Perpignan Via Domitia, France (2) Sorbonne Universités (UPMC, Université Paris 06)-CNRS-IRD-MNHN, LOCEAN Laboratory, 4 place Jussieu, F-75005 Paris, France (3) IFREMER, Centre de Brest, Plouzané, France (4) GEOGLOB, Université de Sfax, Tunisia.

## Abstract

Expanded marine Holocene archives are relatively scarce in the Mediterranean Sea because most of the sediments were trapped in catchment areas during this period. Mud belts are most suitable targets to access expanded Holocene records. These sedimentary bodies represent excellent archives for the study of sea-land interactions and notably the impact of the hydrological activity on sediment accumulation. We retrieved a 7.2 m-long sediment core from the Rhone mud belt in the Gulf of Lions in an area where the average accumulation rate is of ca. 0.70 m/1000 years. This core thus provides a continuous and high-resolution record of the last 10 ka cal BP. A multi-proxy dataset (XRF-core scan, $^{14}$C dates, grain size and organic matter analysis) combined with seismic stratigraphic analysis was used to document decadal to centennial changes of the Rhone hydrological activity. Our results show that 1) the Early Holocene was characterized by high sediment delivery likely indicative of local intense (but short duration) rainfall events, 2) important sediment delivery around 7 ka cal BP presumably related to increased river flux, 3) a progressive increase of continental/marine input during the Mid-Holocene despite increased distance from river outlets due to sea-level rise possibly related to higher atmospheric humidity caused by the southward migration of the storm tracks in the North Atlantic, 4) multi-decadal to centennial humid events in the Late Holocene. Some of these events correspond to the cold periods identified in the North Atlantic (Little Ice Age, LIA; Dark Age) and also coincide with time intervals of major floods in the Northern Alps. Other humid events are also observed during relatively warm periods (Roman Humid Period and Medieval Climate Anomaly).

## 1. Introduction

The Holocene climate is characterized by centennial-scale climate changes that punctuated the final deglacial warming after the Younger Dryas (Renssen et al., 2009; Rogerson et al., 2011; Wanner et al., 2008). Wanner et al. (2014) provided an extensive review of Holocene climate variability mainly based on chronologically well-constrained continental temperature time series that emphasize the superimposition of the insolation-driven climate changes with those induced by other external forcings such as solar activity, volcanism and greenhouse gases ($CH_4$, $CO_2$ and $NO_2$). Based on existing data, Holocene climate can be divided into four periods:

a) the early Holocene (between 11.7 and 8.2 ka cal BP, Walker et al., 2012) characterized by a progressive warming inducing ice-cap melting and outbreaks of freshwater from North America glacial lakes leading to a regional cooling in the Northern Hemisphere, *i.e.* the 8.2 ka cal BP cold event (Barber et al., 1999);

b) the warm middle Holocene (between 8.2 and 4.2 ka cal BP, Walker et al., 2012) that coincides approximately with the Holocene Thermal Maximum (HTM) and is punctuated by several cold relapses (CR) (Wanner et al., 2011). Events at 6.4, 5.3 and 4.2 ka cal BP are the most significant in terms of temperature change (Wanner et al., 2011). The 4.2-ka event corresponds to enhanced dryness in the Southern Mediterranean, Asia and North America, that presumably played a role in the collapse of various civilizations (Magny et al., 2013).

c) the cold late Holocene (from 4.2 ka cal BP to the mid 19[th] century, Walker et al., 2012) that includes the 2.8 ka cal BP cold event possibly responsible for the collapse of the Late Bronze Age civilization (Do Carmo and Sanguinetti, 1999; Weiss, 1982) and the Migration Period cooling around 1.4 ka cal BP (Wanner et al., 2014). The late Holocene cooling trend culminated during the Little Ice Age (LIA) between the 14[th] and 19[th] century (Wanner et al., 2011);

d) the warm Industrial Era from 1850 AD onwards (Rogerson et al., 2011; Wanner et al., 2011).

In contrast to these cool events, the Medieval Climate Anomaly (MCA, 800-1300 AD) is often described as a warm period characterized by intense dryness in some regions of the Northern Hemisphere, such as for example Europe and the Mediterranean region although not synchronous worldwide (PAGES-2k-Consortium, 2013).

The causes of Holocene climate variability are not yet fully understood despite recent advances achieved through the study of climate archives from all around the word from both marine and continental settings. To what extent these well-known climate events are global rather than regional and what are the driving mechanisms at play are still open questions. Numerical modelling allows examining in more details and on a broader geographical scale causes of rapid climate changes and the role of natural or anthropogenic forcings by better integrating data from marine, land and ice archives. Nonetheless, there are significant discrepancies between proxy reconstructions and numerical simulations that suggest the need to generate better chronologically constrained high-resolution proxy records from continental and marine archives and develop new approaches (Anchukaitis and Tierney, 2013; Evans et al., 2013). Of particular interest are the locations that allow developing paleo-hydrological and paleo-environmental investigations at the land / sea interface to better link atmospheric

circulation controlling the precipitation pattern over the continent and changes in the
thermohaline circulation.
Sediment drifts fed by water streams connected to the deep sea such as the Var (Bonneau et
al., 2014) or mid-shelf mud belts are interesting locations to recover sedimentary archives
where both continental and marine proxies can be analyzed. Mid-shelf mud belts, in
particular, are depot centers fed by streams that result from various processes including
diffusion under the influence of storms, advection by currents and transport by gravity flows
(Hill et al., 2007). They often form elongated sediment bodies, between 10-30 m and 60-
100 m water depth, roughly parallel to the coastline. Such sediment bodies can reach several
tenths of meters in thickness when they are associated to large streams, and form infralittoral
prograding prisms (sometimes called subaqueous deltas) as for instance along the Italian
Adriatic coast (Cattaneo et al., 2003). Somehow, they are shallow-water equivalents to
contourites, but they generally display higher accumulation rates making them ideal targets
for paleo-environmental reconstructions.
In this study, we present a continuous record of the Holocene climate obtained from a 7.03 m-
long sediment core retrieved from the Rhone mud belt in the Gulf of Lions. Owing to the high
sedimentation rate of this environmental setting, we could generate sedimentological data at
decadal scale resolution for sediment grain size and semi-quantitative chemical composition
obtained by mean of continuous X-ray fluorescence. Organic matter parameters and the
overall seismic architecture of the mud-belt were also used to reconstruct the terrigenous flux
and the degree of alteration of land-derived material for investigating the relationship between
detritic fluxes and the paleohydrology of peri-Mediterranean rivers. Based on the comparison
of available data, we explored the linkages between rapid climate changes and continental
paleo-hydrology with a focus on the Rhone river flood activity.

## 2. Environmental and climatic framework

### 2.1. The Gulf of Lions geological and oceanographic settings

The Gulf of Lions (GoL) is a passive and prograding continental margin with a relatively
constant subsidence and a high sediment supply (Berné and Gorini, 2005). Located in the
north-west sector of the Mediterranean Sea, the GoL is bounded to the West and to the East
by the Pyrenean and Alpine orogenic belts, and comprises a crescent-shaped continental shelf

with maximum width of 72 km near the mouth of Rhone (Berné et al., 2004). The general oceanic circulation is dominated by the geostrophic Liguro-Provençal or Northern Current (Millot, 1990), which is the northern branch of the general cyclonic circulation in the western Mediterranean basin. This current flows southwestward along the continental slope and temporally intrudes on the continental shelf during northwesterly winds events (Millot, 1990; Petrenko, 2003). Surface water circulation in the GoL shelf is wind-dependent (Millot, 1990). Different wind patterns affect the circulation and transport of suspended particles on the shelf and produce distinctive wave regimes. The continental cold and dry winds known as the *Mistral* and *Tramontane*, blowing from the N and NW through the passages between the Pyrenees, Massif Central and the Alps, are associated with a short fetch that generate small waves on the inner shelf. During winter, these winds induce strong cooling and mixing of the shelf-waters triggering dense water formation (Estournel et al., 2003) and locally generating upwelling (Millot, 1990). Episodic and brief E-SE (*Marin* or Maritime regime) winds are associated with long fetch and large swells. This wind regime induces a rise in sea level along the shore and intense cyclonic circulation on the shelf (Ulses et al., 2008) producing alongshore currents and down-welling (Monaco et al., 1990). Transport of humid marine air masses over the coastal relief induces abundant precipitations often accompanied by river flooding.

The main source of sediment in the GoL is the Rhône River (Pont et al., 2002) and to a lesser extent, small rivers of the Languedoc-Roussillon region (Hérault, Orb, Aude, Agly, Têt ,Tech) (Figure 1). The latter experience episodic discharges (*flash floods* in spring and fall) that are difficult to quantify. The terrigenous sediment supply originating from the Rhone River represents 80% of the total sediment deposited on the shelf (Aloisi et al., 1977). The Rhone River drains a largely mountainous catchment area of 97 800 km² incising a geologically heterogeneous substrate, consisting of siliciclastic and carbonate sedimentary rocks in valley infills and a crystalline (plutonic and metamorphic from the Alpine domain) bedrock. The mean annual water discharge measured at Beaucaire gauging station, downstream the last confluence is 1,701 m$^3$ s$^{-1}$ (mean for 1961-1996); the solid discharge varies between 2 to 20 10$^6$ tons yr$^{-1}$ (Eyrolle et al., 2012; Pont et al., 2002).

Most of the sediment delivered by the Rhone is trapped on the inner shelf, mainly in prodeltas (Fanget et al., 2013; Ulses et al., 2008) but redistribution processes operating along the shelf create mid-shelf depocenters of fine sediments. The sediment accumulation rate varies from 20 to 50 cm yr$^{-1}$ at the present Roustan mouth of the Rhone River and strongly decreases with

the distance from the river. Sediment is exported seaward by several turbid layers: the surface nepheloid layer, related to river plume; an intermediate nepheloid layer that forms during periods of water-column stratification; and a persistent bottom nepheloid layer whose influence decreases from the river mouth to the outer shelf (Calmet and Fernandez, 1990; Naudin et al., 1997). The surficial plume is typically a few meter-thick close to the mouth but rapidly thins seaward to few centimeters (Millot, 1990); it is deflected southwestward by the surface water circulation on the GoL shelf. The predominance of the Rhone River in the sediment supply and the continental shelf circulation allow the identification of several zones in the GoL (Durrieu De Madron et al., 2000): i) the deltaic and prodeltaic sediment units where most of the sediments are trapped, ii) the mid-shelf mud belt between 20 and 50-90 m depth resulting from sediment transport under the influence of the main cyclonic westward circulation and iii) the outer shelf where fine-grained sedimentation is presently very low and where relict fine sands are episodically reworked during extreme meteorological events (Bassetti et al., 2006).

## 2.2. Holocene paleohydrology in the western Mediterranean

The hydrological budget in the Mediterranean borderlands depends on the seasonality of precipitation as well as the catchment geology, vegetation type and geomorphology of the region. In northwestern Mediterranean the most important fluvial discharges occur in spring and autumn, while minimum flow is observed in summer (Thornes et al., 2009). On Holocene time scale, the Mediterranean fluvial hydrology is characterized by the alternation of wet and dry episodes related to changes in atmospheric circulation leading to a North-South hydrological contrast in the Mediterranean region with climate reversal occurring at about 40°N (Magny et al., 2013). Complex climate regimes result from external forcing (orbital, solar activity, volcanism) as well as from internal modes of atmospheric variability such as the North Atlantic Oscillation, East Atlantic, East-Atlantic-West Russian or Scandinavian modes (Josey et al., 2011; Magny et al., 2013).

In the NW Mediterranean, the Holocene fluvial hydrology has been reconstructed using major hydrological events (extreme floods and lake levels) recorded in lake and fluvial sediments (Arnaud et al., 2012; Benito et al., 2015; Magny et al., 2013; Wirth et al., 2013). Overall, the early Holocene climate was generally dry except for short pulses of higher fluvial activity reported in the Durance and southern Alps rivers (Arnaud-Fassetta et al., 2010). A marked cooling trend is observed with a major change around 7,500 a cal BP (Fletcher and Sánchez Goñi, 2008) corresponding to humid conditions in the Iberian peninsula (Benito et al., 2015).

The mid-Holocene (from ca. 7,000 to 5,000 a cal BP) also records low torrential activity but increasing flood frequency between 6,000 and 4,500 a cal BP in Spain, Tunisia and southern France (Arnaud-Fassetta, 2004; Benito et al., 2003; Faust et al., 2004) that evolves in the late Holocene to a general increase of fluvial activity, at least in the Rhone basin catchment and north Alps domain (Wirth et al., 2013). In addition, anthropogenic activities (agriculture and deforestation) over the last 5,000 years have modified the erosional rate in the catchment area, resulting in increased/decreased sediment delivery to the sea depending on the deforestation/forestation phases related to the agricultural development (Arnaud-Fassetta et al., 2000; van der Leeuw, 2005).

## 2.3. Deglacial and Holocene history of the Rhone Delta

During the last ca. 20 ka, the morphology of the Rhone delta strongly evolved in response to sea-level and climate changes. At the end of the Last Glacial Maximum, the Rhone reached the shelf edge and directly fed the Petit Rhone Canyon (Figure 1) (Lombo Tombo et al., 2015). The disconnection between the river and the canyon head is dated at 19 ka cal BP in response to rapid sea-level rise (*ibid.*). The landwards retreat path of the estuary mouth on the shelf has been tracked through the mapping and dating of paleo-delta lobes (Berné et al., 2007; Fanget et al., 2014; Gensous and Tesson, 2003; Jouet, 2007; Lombo Tombo et al., 2015) and, onshore, through the study of ancestral beach ridges (Arnaud-Fassetta, 1998; L'Homer et al., 1981; Vella and Provansal, 2000). During the Younger Dryas, an "Early Rhone Deltaic Complex" (ERDC) formed at depths comprised between -50 and -40 m below present sea level (Berné et al., 2007). The estuary then shifted to the NW as sea-level rose during the Early Holocene (Fanget et al., 2014). The period of maximum flooding in the delta (the turnaround between coastal retrogradation and coastal progradation) is dated at ca. 8,500 -7,500 a cal BP (Arnaud-Fassetta, 1998). Around this time, the mouth of the Rhone was situated about 15 km North of its present position. Between this period and the Roman Age (approximately 20 BC-390 AD in Western Europe), the position of the Rhone outlet(s) are not precisely known and many distributaries, with their associated deltaic lobes, have been identified. However, there is a general consensus on the eastward migration of the delta from the St Ferreol Distributary that occupied the position of the modern Petit Rhone between ca. 6,000 and 2,500 a cal BP, and the modern Grand Rhone, built at the end of the 19[th] century. To the West of the Rhone, a mud belt/subaqueous delta, about 150 km in length, up to 20 m thick, is observed (Figure 1). So far, little attention has been paid to this sediment body, and neither seismic data nor detailed core analysis were available.

## 3. Material and methods

The gravity core KSGC-31, 7.03 m long, was retrieved from the Rhone mud belt (43°0'23''N; 3°17'56''E, water depth 60 m) during the GM02-Carnac cruise in 2002 on the R/V "Le Suroît". Seismic data were acquired in 2015 aboard R/V Néréis during the Madho1 cruise, using an SIG$^{TM}$ sparker. The shooting rate was 1s. Data were loaded on a Kingdom$^{TM}$ workstation. An average seismic velocity of 1,550 ms$^{-1}$ (based on measurements of sonic velocity with a Geotek$^{TM}$ core logger) was used to position the core data on seismic profiles. The uncertainty in the position of time lines on the seismic profile at the core position is on the order of $\pm$ 0.5 m, taking into account the resolution of the seismic source, the errors in positioning and sound velocity calculation. Due to the shallow water depth, core deformation by cable stretching is considered as negligible.

Grain size analyses were carried out by mean of a Malvern$^{TM}$ Mastersize 3000 laser diffraction particle size analyzer using a HydroEV dispersing module, which measures particle grain-sizes between 0.04 and 3,000 µm. Samples were dispersed in a solution of $(NaPO_3)_6$ (1.5 gr/l of distilled water) for 1 hour in order to better disaggregate the sediment. Before each measurement, the sample was stirred on a rotating mixer during 20 minutes. Grain-size parameters were measured all along the core every cm. Three size ranges were used to classify the grains: clay (<8 µm, as recommended by Konert and Vandenberghe (1997), coarse silt (>8 µm and <63 µm), sand (>63 µm and < 250 µm). The D50, representing the maximum diameter of 50% of the sediment sample was calculated.

Core KSGC31 was analyzed using an Avaatech XRF Core Scanner at IFREMER (Brest, France). This non-destructive method provides semi-quantitative analyses of major and minor elements by scanning split sediment cores (Richter et al., 2006). Measurements were performed every 1 cm with a counting time of 20 sec and a 10kV and 30kV acceleration intensity. Resulting element abundances are expressed as element-to-element ratio. Three ratios are used in this work: 1) **Ca/Ti** ratio, to account for two end-members in the sediment composition. The Ca is supposedly mostly derived from biogenic carbonates, while Ti is commonly used for tracking terrigenous sediments, even if usually found in small amounts. Nonetheless, it is worthwhile reminding that calcite of detritic origin, generated by erosion of calcareous massifs in the catchment area, represents an important component of the fluvial Rhône waters sediment. This type of calcite is transported into the sea but it is mainly accumulated in the sand fraction, trapped in the proximal deltaic sediments. In the mud belt where deposits are mostly pelitic, the detritic calcite quickly decreases seaward of the river

mouth, with only a very small fraction being preserved in the clay fraction (Chamley, 1971). On the other hand, calcite of biogenic marine origin (bioclasts) is usually abundant. Benthic (rare planktonic) foraminifera, ostracods, fragmented mollusk shells and debris from bryozoan and echinoids can be observed under the binocular microscope. Thus, the Ca content in the core KSGC31 is considered as related to biogenic marine productivity; 2) **Zr/Rb** reflects changes in grain size, with higher values in the relatively coarse grained sediments. Zr is enriched in heavy minerals and commonly associated with the relatively coarse-grained (silt-sand) sediments fraction (highest Zr values are found in sandstones), whereas Rb is associated with the fine-grained fraction, including clay minerals and micas (Dypvik and Harris, 2001) ; 3) **K/Ti** values can be related to illite content. Illite is formed by weathering of K-feldspars under subaerial conditions and most of the K leached from the rocks is adsorbed by the clay minerals and organic material before it reaches the ocean (Weaver, 1967). In the case of the GoL, the Rhône waters deliver mainly illite and chlorite to the Mediterranean Sea whereas rivers flowing from Massif Central, Corbières and Pyrénées mainly carry illite and montmorillonite (Chamley, 1971). Thus, illite (K) is thought to be abundant in fluvial waters ending in the GoL, and thus K relative abundances can be used as a proxy for sediment continental provenance. Because illite might be depleted in K upon pedogenetic processes, the K/Ti ratio can be considered as an indirect proxy for the intensity of chemical weathering (Arnaud et al., 2012).

The XRF raw data were smoothed using a 5-point moving average to remove background noise.

In addition, semi-quantitative bulk geochemical parameters such as total carbon (TC), organic carbon (OC) and total nitrogen (TN) were determined from freezed-dried homogenized and precisely weighed sub-samples of sediment using the Elementar Vario MAX CN automatic elemental analyzer. Prior to the OC analyses, samples were acidified with 2M HCl overnight at 50°C in order to remove carbonates (Cauwet et al., 1990). The precision of TC, OC and TN measurements was 5 and 10%. The calcium carbonate content of the sediments was calculated from TC-OC using the molecular mass ratio ($CaCO_3$: C= 100:12). Results are expressed as the weight percent of dry sediment (% d.w.). The atomic C:N ratio ($C:N_a$) was calculated and used as a qualitative descriptor of organic matter (OM). Moloney and Field (1991a) proposed $C:N_a = 6$ for OM of marine origin because of the high protein content of organisms such as phytoplankton and zooplankton. Higher plant-derived OM of terrestrial origin have higher $C:N_a$ ratios (>20) than marine organisms because of a high percentage of non-protein constituents (Meyers and Ishiwatari, 1993). In marine sediment, $C:N_a$ ratios are usually higher

than phytoplankton. C:N$_a$ ratios comprised between 6 and 10 are indicative of degraded organic detritus resulting from the breakdown of the more labile nitrogenous compounds and values of C:N$_a$ ratio > 13 indicate a significant contribution of terrestrial organic matter (Goñi et al., 2003).

The age model is based on 21 radiocarbon dates (Table 2) obtained by Accelerator Mass Spectrometry (AMS) at the Laboratoire de Mesure du Carbone 14, Saclay (France). The two uppermost dates were performed at Beta Analytic Radiocarbon Dating Laboratory and indicate post-bomb values (AD 1950). The [14]C dates were converted into 1σ calendar years using Calib7.1 (Stuiver and Reimer, 1993) and the MARINE 13 calibration dataset including the global marine reservoir age (400 years) (Charmasson et al., 1998). We used a local marine reservoir age correction of ΔR = 23 ± 71 years (http://calib.qub.ac.uk/marine/regioncalc.php). The age model was obtained by polynomial interpolation between [14]C dates excluding the minor reversal at 18.5 cm (350 ± 78 yrs) and the two post-bomb dates. Timing and uncertainty for the main events is estimated using the Bayesian approach of OxCal 4.2 (Ramsey and Lee, 2013) (Tables 3). We used the same age model as in Jalali et al. (2016). Age inversions are not used in the estimation of the sedimentation rate (SR) (Table 2).

## 4. Results

### 4.1. Age model, sedimentological core description

Core KSGC31 was retrieved at the seaward edge of the Rhone mud belt. The seismic profile at the position of the core displays the architecture of this mud belt that drapes Pliocene rocks and continental deposits of the Last Glacial Maximum (Figure 2). The bottom of the core corresponds to the *ravinement* surface (RS in Figure 2) that formed by wave erosion at the time of marine flooding during the deglacial period. This 20 cm-thick heterolitic interval includes fluvial and coastal sands and gravels mixed with marine shells in a muddy matrix. At the position of core KSGC31, it is postdated by the overlying muds immediately above (ca. 10,000 a cal BP). The period of "turn around" between coastal retrogradation and coastal progradation is well marked on the seismic profile by a downlap surface dated at ca. 7.5 ka cal BP at the position of the core. It corresponds to the Maximum Flooding Surface in the sense of Posamantier and Allen (1999). Two other distinct seismic surfaces (higher amplitude, slightly erosional) can be recognized in the upper part of the wedge (Figure 2), they are dated at ca. 4.2 and 2.5 ka cal BP from the core.

Based on the 21 [14]C dates, the average SR has been estimated to ~0.70 m/1,000 years. The absolute chronology allows identifying three stratigraphic intervals corresponding to the formal subdivision of the Holocene epoch proposed by (Walker et al., 2012). The well-known cold events (Cold Relapses, CRs) are defined on the basis of this chronology (Figure 3, Table 1) and used in this paper to highlight possible correlation with local conditions.

The core is predominantly composed of silt (60-70%) and clay. The clay content is highly variable but no more than 50% between 10,000 and 4,000 a cal BP, and between 50 and 60% in its upper 350 cm corresponding to the last 4,000 years (Figure 3). Small-size shell debris are randomly mixed with the clayey silt but become more abundant between 400 and 500 cm depth. Abundant and well-preserved *Turritella* sp shells certainly not reworked are found between 680 and 640 cm. The sand fraction is generally very low (0.5-5%) except for the lowermost 30 cm (50%, Figure 3). At visual inspection, the thin sandy base (between 703 and 690 cm) contains very abundant shell debris. Weak bioturbation is visible on the X-ray images as well as the occurrence of sparse articulated shells.

### 4.2. Elemental and geochemical distribution

Ca/Ti, K/Ti and Zr/Rb ratios were generated and cross-analyzed with grain-size (clay content -D50 computed curve) and C:$N_a$ to assess changes in geochemical composition.

**In the Early Holocene,** the Ca/Ti ratio is fairly constant and relatively high. The carbonate content is high (>45% $CaCO_3$, Figure 4b), whereas C:$N_a$ values are highly fluctuating between values of 20 (~ 10 ka) and lower values of 13 towards the mid-Holocene (Figure 4a). Zr/Rb ratios gradually decrease while K/Ti shows a relatively stable behavior. Between 7,000 and 9,000 a cal BP, K/Ti and Zr/Rb indicate lower values, yet with a peak in the mid-interval, around 8,200-8,300 a cal BP (Figure 4d,e). All over the period, clay content is comprised approximately between 24 and 52% (Figure 5c), D50 is generally >10 µm and variable (Figure 4f). A significant drop of Ca/Ti and D50 is observed in the 7,000-6,400 a cal BP interval (Figure 4c, f). Similar trends are observed for the K/Ti and Zr/Rb, but the most abrupt drop occurs between 6,500 and 6,400 a cal BP. No significant changes are detected in the main lithology (mostly clayey, Figure 3). C:$N_a$ ratios decrease (<13) due to a better preservation of nitrogen in clay deposits.

**After 6.4 ka cal BP**, Ca/Ti displays a constant decreasing trend until 4,200 a cal BP. On the other hand, C:$N_a$ between 6,400 and 4,200 a cal BP reveals two prominent peaks (>15) culminating at 5,700 and 4,800 a cal BP (Figure 4a) that roughly correspond to low K/Ti and Zr/Rb values (Figure 4d, e), higher clay (Figure 5c) and lower carbonate sediment contents

(Figure 4b). The most pronounced changes in the elemental ratio are observed after 4,200 a cal BP (Figures 4 and 5). Millennial-scale oscillations are discernible in the Ca/Ti record (Figure 4c) and coherent with changes in K/Ti and Zr/Rb ratios (Figure 4d, e) and, to some extent, with the D50 values (Figure 4f). Six main episodes of high terrigenous inputs (lowest Ca/Ti) are clearly expressed in the XRF data at ~3,500 ± 170, ~2,840 ± 172, ~2,200 ± 145, ~1,500 ± 124, ~1,010 ± 75 and ~720 ± 72 a cal BP (Figure 4, Table 3). Considering the age uncertainty, only some of those events might coincide with CRs (CR6, CR5, CR4, Figure 4). The peaks in the clay content correspond to low Ca/Ti ratios of variable amplitude. The clay content of the 2,840 and 2,200 a cal BP events are among the highest (~35%, Figure 5e). From 4,200 a cal BP to present, the $C:N_a$ values decrease gradually. Between ~4,200 and ~3,200 a cal BP, some values exceed 13. Thereafter, the $C:N_a$ values range between 9 and 10 (Figure 4a). The Late Holocene is also characterized by decreasing carbonate content with a drastic drop around 2,000 a cal BP (Figure 4b). The SR is also higher than during the Mid-Holocene lying between 0.5 and 1 mm/year.

## 5. Interpretation and Discussion

Numerous forcing factors (*i.e.* sea-level, ice cap extent, forest cover, volcanic activity, etc.) may account for the climate variability in the Holocene. Statistical analysis of proxy time series in both northern and southern hemisphere (Wanner et al., 2011) have demonstrated that multidecadal to multicentury cold relapses (CRs) interrupted periods of relative stable climate conditions. They are demonstrated to exist at least in the North Atlantic (Bond, 1997) and surrounding land areas. However, there is a general agreement about the different local expressions and timing offset of these rapid climate changes according to geographical position or geomorphological setting. In a way, these events cannot be considered as really global, but they nonetheless represent significant milestones in the Holocene climate history. In this paper, we use the correlation with CRs known from the literature (Table 3) in order to highlight possible differences in features and chronology of rapid events between Atlantic and western Mediterranean during Early, Middle and Late Holocene.

*Early Holocene (11.7-8.2 ka cal BP)*

The lower 20 cm of the core are made of heterolithic coarse-grained sediments of continental origin mixed with abundant shell debris. This interval corresponds to the *ravinement* surface seen on seismic profiles; it formed by transgressive erosion when relative sea-level was -30/40 m lower than today. It is unconformably overlaid by fine-grained sediments that

represent the initiation of the mud belt, around 9,000 a cal BP. The ~9-8.2 ka interval is marked by highest SR values and high terrestrial supply, as also indicated by the high $C:N_a$ ratio (>13) (Figure 4a) (Buscail and Germain, 1997; Buscail et al., 1990; Gordon and Goñi, 2003; Kim et al., 2006). Of note, the $C:N_a$ ratios ~ 20 indicative of even larger enrichment in organic material originating from soils or plant debris in the coarse deposit at the very bottom of the core (700 cm) (Hedges and Oades, 1997; Meyers and Ishiwatari, 1993). A layer of high *Turritella* abundances is identified in the fine-grained sediments just above the sandy interval (680-640 cm, i.e. 8,500 -8,000 a cal BP) (Figure 3). Then *Turritella* shells disappear gradually towards the top of the core, suggesting an upward deepening environment. The high *Turritella* level could indicate a change in Northern Hemisphere climate and can be hypothetically related to the "*Turritella* Layer" described by Naughton et al. (2007) on the NW Atlantic shelf, therefore suggesting a regional change between 8,700 and 8,400 a cal BP, possibly in relation with the southward migration of the Boreal biogeographical zone. The Maximum Flooding Surface (MFS) is dated around 7,500 a cal BP (Figure 2). This age may vary at different locations because it depends upon the ratio between sediment delivery and accommodation space, but it matches well the age of delta initiations observed worldwide by Stanley and Warne (1994).

The increase of K/Ti between 9,000 and 7,000 a cal BP might reflect the gradual decrease of the contribution of weathered material from the river catchment areas, which can thus be interpreted as a signal of weaker pedogenetic processes and lower soil erosion due to dry climate in European Alps (Figure 4d) (Arnaud et al., 2012).

The period between ~ 12,000 and 7,000 a cal BP is marked by a continuous retreat of Arctic continental ice-sheets until the complete disappearance of the Fennoscandian and Laurentide ice cap (Tornqvist and Hijma, 2012; Ullman et al., 2015). Ice sheet melting is seen in the general sea-level rise and also manifested by short-lived water releases into the ocean and occasionally perturbing the North Atlantic Ocean circulation and climate over Europe (for example the 8,200 cal BP event, here CR0). It is worthwhile to note that around CR0, the K/Ti ratio shows a peak within an interval of low values between approximately 7,900 and 8,300 a cal BP. This peak would identify an increase of continental supply and low chemical weathering corresponding to cold (weak soil formation) and wet (high physical erosion) conditions over mid-latitude Europe in response to the 8,200 a cal BP cooling (Arnaud et al., 2012; Magny et al., 2003). This phenomenon is also attested by higher lake levels in Western Europe (Figure 5d, f) concurrent with CR0 (Magny et al., 2013). Note that no clear

temperature drop in the alkenone-derived SST record generated in the core has been detected
(Jalali et al., 2016).

*Mid-Holocene (8.2-4.2 ka cal BP)*
Values of $C:N_a$ ratios are mainly >13 (Fig. 4a) between 6.3 and 4.4 ka cal BP, while Ca/Ti
shows a slight progressive decrease that can be interpreted as an increase of terrestrial inputs
during the Mid-Holocene, despite increasing distance of KSGC31 site from river outlets due
to sea-level rise and the progressive shift of the Rhone delta to the East (Fanget et al., 2014).
The Mid-Holocene is described as a period of relatively mild and high atmospheric moisture
balance (Cheddadi et al., 1998) that favored the maximum expansion of the mesophytic forest
leading to a maximum land cover over Europe. Nonetheless, two main short-lived climate
anomalies are reported at 6,600 -5,700 a cal BP (CR1, Tables 1 and 3) and 5,300 - 5,000 a cal
BP (CR2, Tables 1 and 3) over the North Atlantic (Wanner et al., 2011) and in Europe
(Magny and Haas, 2004; Robert et al., 2011), at the time of global cooling. CR1 is associated
with drying climate in eastern Europe and Asia and has been related to the weakening of the
Asian monsoon and the decrease of summer insolation (Gasse et al., 1991), while CR2
coincides with weaker solar activity as indicated by maximum atmospheric [14]C around 5,600–
5,200 a cal BP (Stuiver et al., 2006), lower tree lines (Magny and Haas, 2004) and colder sea-
surface temperatures (Jalali et al., 2016).
According to our data, during CR1 and CR2, chemical weathering was weak as suggested by
high K/Ti values (Figure 4d), mean SR was generally low (<1 mm/yr on average, Table 2) but
there was no significant change in terrigenous inputs (Ca/Ti ratio, Figure 4c) . The $C:N_a$ ratios
indicate better preservation of nitrogen organic compounds preferentially adsorbed in the clay
fraction (Figure 4a). The reduction of the vegetation cover in the river catchment, combined
with lower (1-1.5°C) temperatures in the European Alps (Haas et al., 1998), may explain the
low chemical degradation state of the illite minerals. A drop in sea surface temperature is also
recorded by alkenones in the core as illustrated in Jalali et al. (2016), confirming the impact of
the cold relapses in the Mediterranean area in the Mid- Holocene (Figure 4i).


*Late Holocene (4.2-0 ka cal BP)*
Multi-decadal to century-scale wet episodes are evidenced from ~ 4,200 a cal BP that marks
the Mid-Late Holocene transition (Figure 3). In the KSGC31 core, wetter intervals are

expressed by highly fluctuating Ca/Ti, Zr/Rb and K/Ti ratios, $C:N_a$ and grain size values (Figure 4 and Table 3). From present to 3.5 ka cal BP, the $C:N_a$ ratio shows an increase from 9 to 11 (Figure 4) testifying active diagenetic processes, due to preferential degradation of nitrogen relative to carbon during burial. Some values > 13 are still observed between 4,200 and 3,500 a cal BP, indicating enhanced terrestrial inputs. Episodes of enhanced terrigenous inputs (during floods, for instance) are detected by low Ca/Ti ratios that also coincide with low Zr/Rb and low D50 values, indicating general smaller-size terrigenous grains as also suggested by high clay content (Figure 5c). Indeed, after the stabilization of sea-level, only the finest sediment fraction (clay) transported by the river plume reaches the mud belt at the core site.

An exception to this pattern is observed for the LIA, when quite high Ca/Ti would suggest relatively "dry" conditions (Figure 4c,f). The qualitative observation under the binocular microscope of the coarse (>63 µm) fraction reveals the presence (only in this specific interval) of abundant bryozoans and *Elphidium crispum* (coastal benthic foraminifer) tests together with rare grains of quartz. The biogenic debris can explain the high Ca content and presence of quartz grains, the peak of Zr/Rb (Figure 4e). The accumulation of this material is maybe due to concomitant occurrence of river floods (Figure 5d) and storms, which might have remobilized coarse material from coastal setting (Bourrin et al., 2015).

Thus, intensified hydrological activity associated with high terrestrial inputs would have prevailed during the Late Holocene, as also suggested by higher SR (Table 2). The enhanced terrestrial inputs are inferred from XRF ratios and discussed in this work, but the biomarker data in Jalali et al. (2016) also highlighted enhanced flood activity during the Late Holocene. The TERR-alkane concentrations are among the highest of the entire Holocene record and with maxima recorded during Common Era (last 2000 years).

A similar signature of continental runoff in marine sediments (low Ca/Ti ratio) during the past ~ 6,500 years has been reported in the central Mediterranean and related to climatically driven wet periods (Goudeau et al., 2014). In the KSGC31, these events (~2840 a, ~ 1500 and ~720 a cal BP, Figure 4) barely coincide with the cold events in the North Atlantic but are concomitant with periods of increasing flood frequency in the Northern Alps as reconstructed by Wirth et al. (2013) (Figure 5d) and punctuated overall warm (and dry) periods such as the MCA at ~ 3,500; ~ 2,200 and ~1,000 and ~0,72 a cal BP (Figure 4). This pattern suggest different causes for enhanced precipitations in the late Holocene.

A possible control of North Atlantic Oscillation (NAO) on the amount of precipitation in the Mediterranean land areas might be put forward. The NAO exerts a strong influence on the

precipitation pattern in Europe and the NW Mediterranean region. Today, precipitation in the western Mediterranean region and southern France is lower during positive NAO. Rainfall increases under negative NAO due to the southern shift of the Atlantic storm tracks leading to enhanced cyclogenesis in the Mediterranean Sea (Trigo et al., 2000). The position of the ITCZ is also important in the precipitation pattern of the Mediterranean region and its southernmost position is the probable cause for extremely dry conditions between 2,500 and 2,000 a cal BP (Schimmelpfennig et al., 2012). The reconstructed NAO index (Olsen et al., 2012) indicates a predominance of positive states between 5,000-4,500 and 2,000-550 a cal BP (Figure 4h), in agreement with a) an increased frequency of floods in Northern Alps (Figure 5d; Wirth et al., 2013), b) higher lake levels at Accesa (Central Italy), Ledro (Northern Italy) and in Central-Western Europe (Magny et al., 2013), all together suggesting more humid conditions in west-central Europe (Figure 5d,f).

Late Holocene human settlements along the Rhone valley and South France also may have had an impact on the origin and amounts of eroded sediments in the river catchment areas. When examining the chemical signature of KSGC-31 sediments, low K/Ti ratios co-eval with wet events (Figure 4d) reflecting soil weathering due to terrain degradation, that could be interpreted as the result of widespread deforestation by agropastoral activities in this area since the end of the Neolithic. The most extensive erosion episodes in the Rhone valley correspond to 1) the end of Neolithic (~ 4,000 BC; 6,000 a cal BP) after the first phase of human expansion linked to the development of the agriculture 2) the end of the Bronze Age (~ 2,000 BC; 4,000 a cal BP) and 3) the Roman Period when a rapid transformation of landscape is operated by deforestation and their replacement by intensively cultivated agricultural land (van der Leeuw, 2005).

Disentangling human impact from climate control on environmental changes in the Late Holocene is not an easy task, and requires the study of other river catchment basins to confirm the regional character of these observations. However, assuming that climate variability is the major factor influencing soil pedogenesis, we can hypothesize that the elemental composition of marine sediments reflects continental erosion and transport because of a good correspondence with temperature variability in the Mediterranean Sea along the same core (Jalali et al., 2016), and because, on a regional scale, both marine and continental climate proxies indicate co-eval signals (Arnaud et al., 2012; Goudeau et al., 2014) . Despite the fact that the characteristics in amplitude and duration of these climate intervals slightly differ geographically, there seems to be a general agreement on their origin and the role of solar forcing and large-scale atmospheric circulation. However, amplification of soil degradation

following waves of human occupation should be further explored through accurate correlation between archeological data and paleoenvironmental proxies in order to better evaluate the importance of land use on sedimentary signals.

**6. Conclusions**

This work represents the first attempt to detect and decipher the linkages between rapid climate changes and continental paleo-hydrology in the NW Mediterranean shallow marine setting during the Holocene.

Based on the combination of sedimentological and geochemical proxies we could demonstrate that between 11 and 4 ka cal BP, terrigenous input broadly increased. A *Turritella*-rich interval is observed in the 8,5-8 ka cal BP interval, which could correspond to a change in Northern Hemisphere climate and can be correlated to the "*Turritella* Layer" described in the NW Atlantic shelf, possibly in relation with the southward migration of the Boreal biogeographical zone.

From ca. 4,000 a cal BP to present, the sediment flux proxies indicate enhanced variability in the amount of land-derived material delivered to the Mediterranean by the Rhone River input. We suggest that this late Holocene variability is due to changes in large-scale atmospheric circulation and rainfall patterns in Western Europe including the increased variability of extension and retreat of Alpine glaciers. Anthropogenic impacts such as deforestation, resulting in higher sediment flux into the Gulf of Lions, are also likely and should be better taken into account in the future.

**Acknowledgements**

We thank MISTRALS/PALEOMEX for financial support and the crew operating the GMO2 Carnac (R/V "Le Suroît") and GolHo (R/V "Néréis") cruises. Nabil Sultan and the crew and science parties aboard R/V Suroit (IFREMER) retrieved core KSGC31 during the GMO2-Carnac cruise. The Captain and crew of R/V Néréis (Observatoire Océanologique de Banyuls), as well as Olivier Raynal and Raphael Certain (CEFREM) are thanked for their assistance during the MADHO 1 cruise. ARTEMIS (Saclay, France) program is acknowledged for performing the [14]C measurements. Two anonymous reviewers are

529     acknowledged for providing suggestions that allowed to improve the quality of the

530     manuscript. S. Luening is thanked for commenting the manuscript during the open discussion.

531

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

**Tables**
Table 1: Chronology of Holocene cold relapses (CR) based on existing literature.

| Event | Time slice (ky) | References |
|-------|-----------------|------------|
| CR0 | 8.2 | (Barber et al., 1999) |
| CR1 | 6.4-6.2 | (Wanner et al., 2011) |
| CR2 | 5.3-5.0 | (Magny and Haas, 2004; Roberts et al., 2011a) |
| CR3 | 4.2-3.9 | (Walker et al., 2012) |
| CR4 | 2.8-3.1 | (Chambers et al., 2007; Swindles et al., 2007) |
| CR5 | 1.45-1.65 | (Wanner et al., 2011) |
| CR6 | 0.55-0.15 | (Wanner et al., 2011) |




Table 2: $^{14}$C dates performed on core KSGC31

| Depth (cm) | Material | Laboratory | Radiocarbon age ±1σ error (yr BP) | Calibrated Age (cal BP) | ± 1σ error | Sedimentation Rate (SR) (mm/year) |
|---|---|---|---|---|---|---|
| 5.5 | *Bittium* sp. | Beta Analytics | 420± 30 | 24[a] | 60 | - |
| 11.5 | *Tellina* sp. | Beta Analytics | 430±30 | 34[a] | 60 | - |
| 18.5 | *Pecten* sp. | Beta Analytics | 720 ± 40 | 350[b] | 78 | - |
| 25.5 | *Venus* sp. | LMC14 | 640 ± 30 | 234 | 99 | 1,34 |
| 41 | *Pecten* sp. | LMC14 | 700 ± 30 | 339 | 79 | 1,48 |
| 52 | Indet. bivalve | LMC14 | 960 ± 30 | 551 | 59 | 0,52 |
| 71 | *Arca tetragona* | LMC14 | 1340 ± 30 | 851 | 80 | 0,63 |
| 110.5 | *Venus* sp. | LMC14 | 1465 ± 30 | 992 | 85 | 2,80 |
| 186.5 | *Nucula* sp. | LMC14 | 2235 ± 40 | 1805 | 99 | 0,93 |
| 251 | Juvenile bivalve shells (ind.) | LMC14 | 2940 ± 30 | 2674 | 100 | 0,74 |
| 330.5 | *Venus cosina* | LMC14 | 3870 ± 30 | 3796 | 106 | 0,71 |
| 370.5 | *Nuculana* sp. | LMC14 | 4170 ± 30 | 4223 | 113 | 0,94 |
| 390.5 | *Turritella* sp. | LMC14 | 4500 ± 30 | 4676 | 106 | 0,44 |
| 460 | *Venus* sp. | LMC14 | 5530 ± 45 | 5873 | 106 | 0,58 |
| 481 | *Ostrea* sp | LMC14 | 5955 ± 35 | 6348 | 78 | 0,44 |
| 501.5 | *Turritella* sp. | LMC14 | 6380 ± 50 | 6826 | 107 | 0,43 |
| 552 | Shells (mixed) | LMC14 | 7215 ± 30 | 7653 | 75 | 0,61 |
| 583 | *Turritella* sp. | LMC14 | 7860 ± 60 | 8288 | 92 | 0,49 |
| 652 | *Turritella* sp. | LMC14 | 8310 ± 35 | 8843 | 121 | 1,24 |
| 700.5 | *Turritella* sp. | LMC14 | 9215 ± 30 | 10006 | 123 | 0,42 |
| 701 | *Turritella* sp. | LMC14 | 9190 ± 50 | 9968 | 145 | - |


Table 3: Time uncertainty (1σ) of "wet spells" identified on the Ca/Ti and K/Ti ratios
(Figure 4)

| Abbreviation in the text | Start (yr cal BP) | Maximum (yr cal BP) | ±1 σ uncertainty | End (a cal BP) | Proxy |
|---|---|---|---|---|---|
| 0.72 ka | 645 | 720 | 72 | 800 | Ca/Ti |
| 1.01 ka | 1000 | 1015 | 75 | 1,070 | Ca/Ti |
| 1.5 ka | 1400 | 1500 | 124 | 1,640 | Ca/Ti |
| 2.2 ka | 2080 | 2200 | 145 | 2,300 | Ca/Ti |
| 2.84 ka | 2700 | 2840 | 172 | 2,900 | Ca/Ti |
| 3.5 ka | 3350 | 3500 | 170 | 3,615 | Ca/Ti |
| 4.8 ka | 4670 | 4800 | 150 | 4,960 | K/Ti |
| 5.7 ka | 5530 | 5700 | 162 | 5,770 | K/Ti |


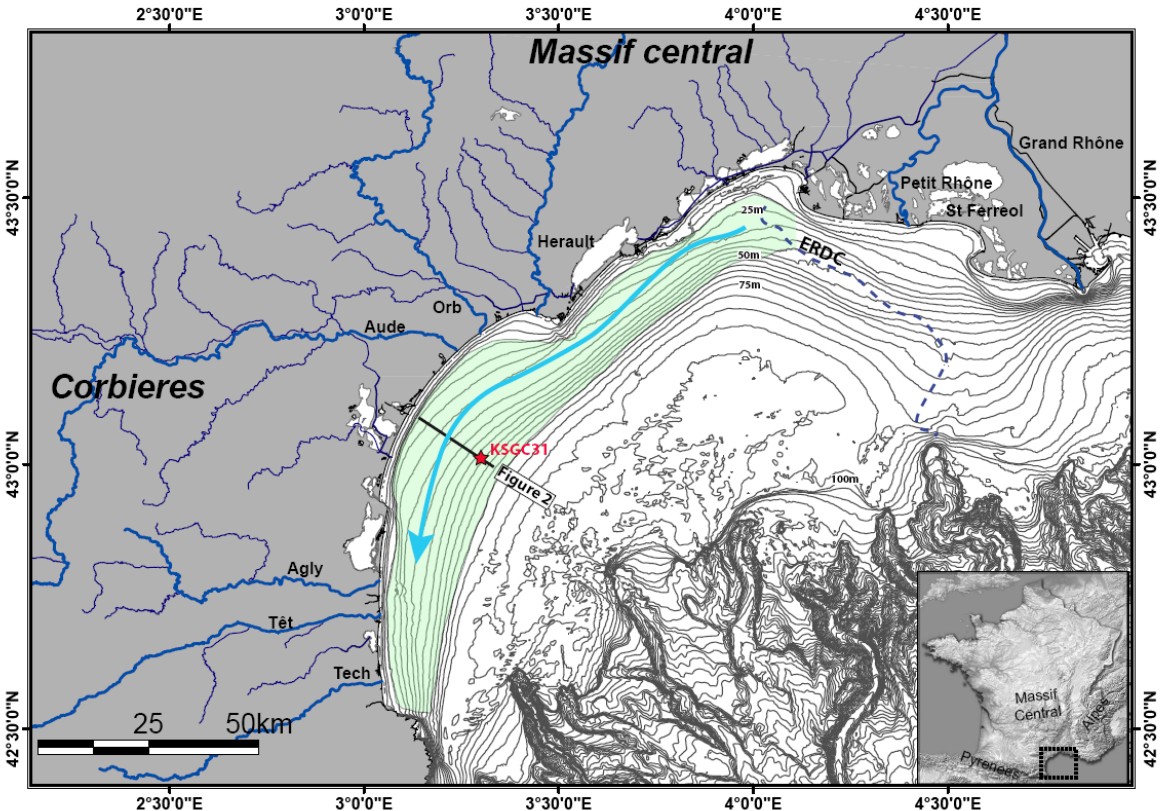

**Figure 1:** Bathymetric map of the Gulf of Lions and position of core KSGC31. The approximate extent of the Rhone mud belt is represented in green; the arrow represents the direction of dominant transport of suspended sediments. Bathymetric map based on Berné et al. (2007). Contour lines every 5 m on the shelf. The dotted line corresponds to the retreat path of the Rhone during the Deglacial (based on Gensous and Tesson, 2003; Berné et al., 2007; Jouet, 2007; Fanget et al., 2014, Lombo Tombo et al., 2015). ERDC: Early Rhone Deltaic Complex.

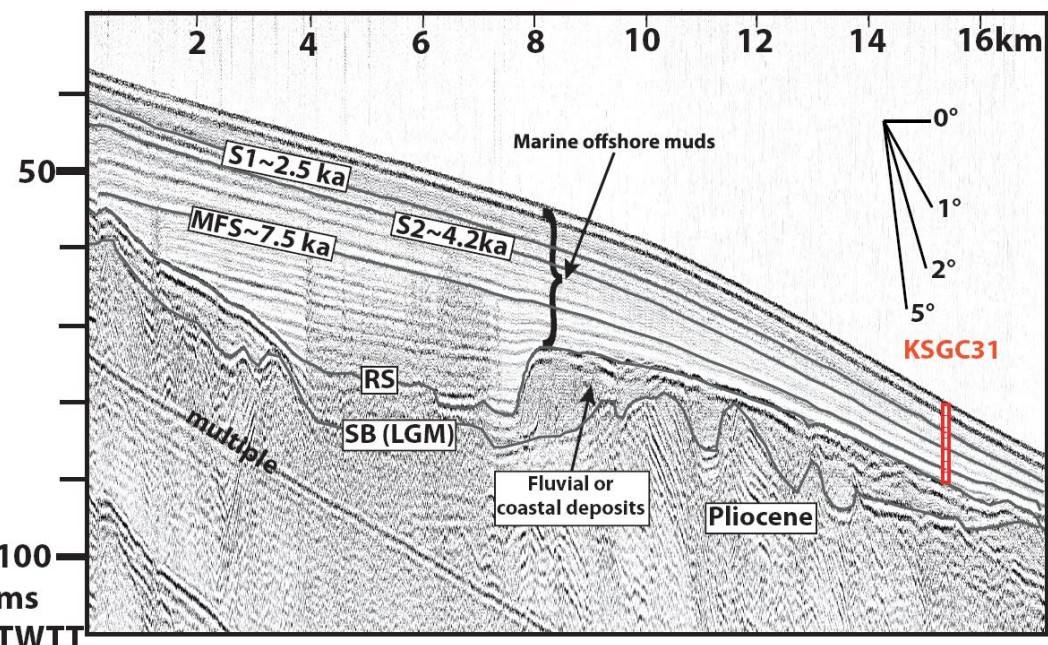

782

**Figure 2:** Seismic profile across the Rhone Mud Belt at the position of core KSGC31 (position in Figure 1). **SB**: Sequence Boundary- surface formed by continental erosion during the Last Glacial Maximum (LGM**). RS**: *Ravinement* Surface formed by wave erosion during sea-level rise. It corresponds to the coarse interval at the base of the core. **MFS** : Maximum Flooding Surface (MFS). It corresponds to the phase of transition between coastal retrogradation and coastal progradation. It is dated here at ca. 7.5 ka cal BP (i.e. the period of global sea-level stabilization). S1 and S2 are seismic surfaces used as time lines (on the basis of the age model in Figure 3 (respectively 4.2 ± 0.5 ka cal BP and 2.5 ± 0.5 ka cal BP). Horizontal bars every meter along the core.

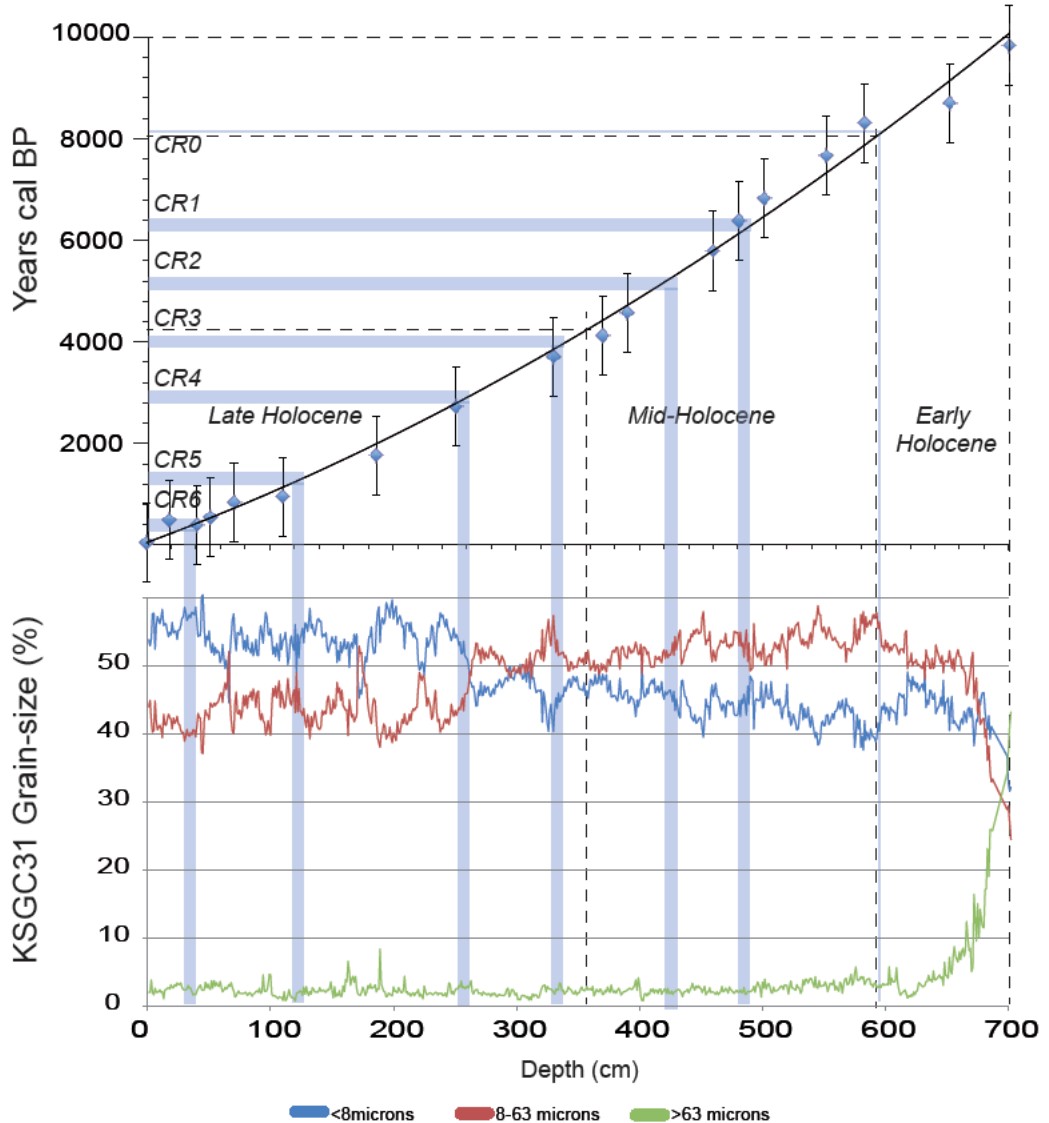


**Figure 3:** Correlation age-depth in core KSGC31. The Holocene time is divided into Early, Middle and late Holocene according to Walker et al. (2012). General core lithology is shown through distribution of three grain-size classes: <8µm, 8-63 µm, >63 µm.





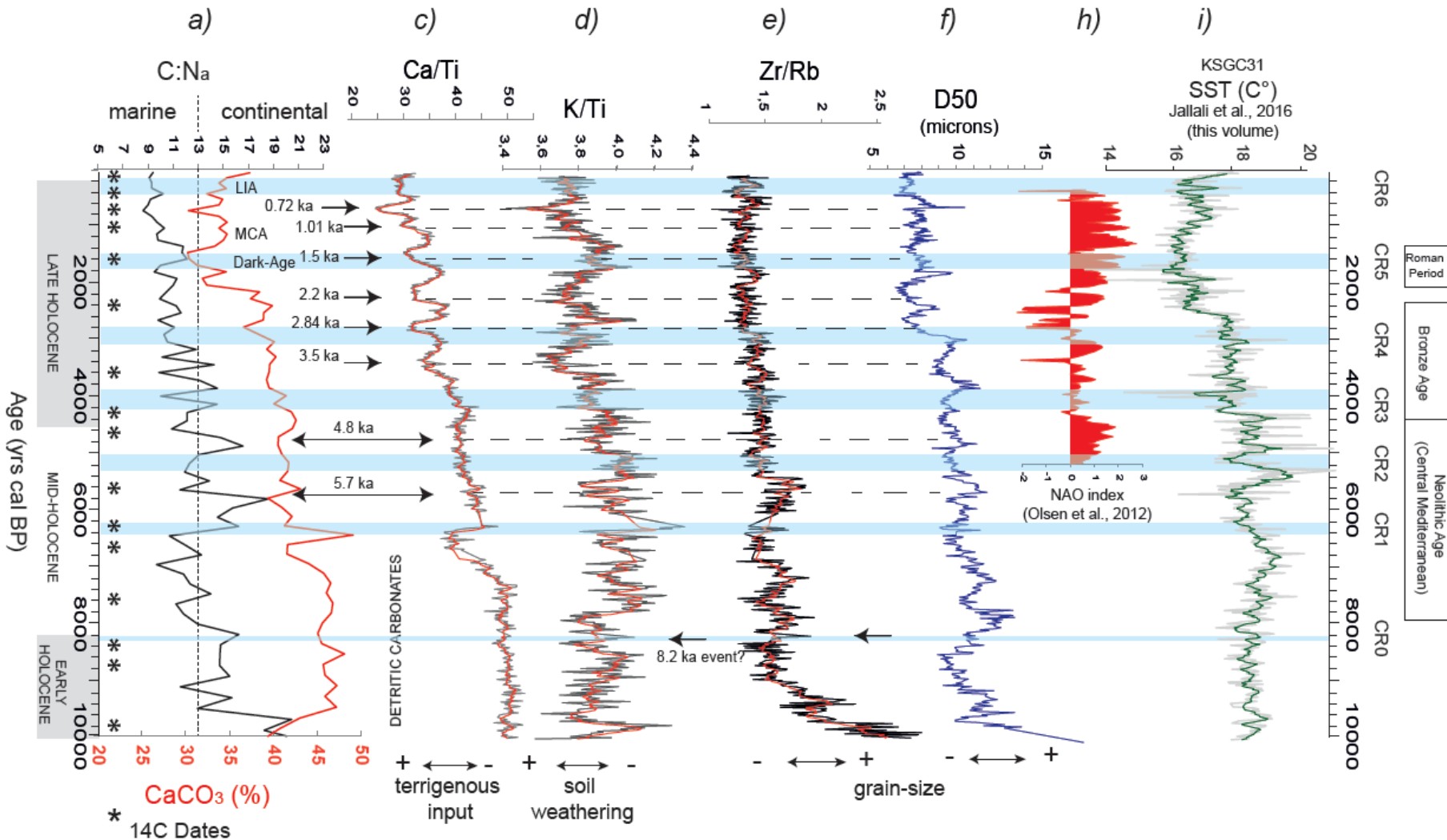

**Figure 4:** KSGC31 geochemical and sedimentological proxies: (a) C:$N_a$ atomic ratio is used as qualitative descriptor of organic matter nature. Values of C:$N_a$ ratio > 13 indicate significant amount of terrestrial organic matter, according to Goñi et al. (2003); (b) $CaCO_3$ content (%) calculated from TC-OC using the molecular mass ratio ($CaCO_3$: C= 100:12); (c) Ca/Ti ratio is used for estimating the degree of detritism, since Ti is commonly found in terrigenous sediments; (d) K/Ti ratio can be related to illite content, formed by weathering of K-feldspars. Illite might be depleted in K upon pedogenetic processes, the K/Ti ratio can be considered as an indirect proxy for the intensity of chemical weathering (Arnaud et al., 2012); (e) Zr/Rb is known to reflect changes in grain size; Zr is commonly associated with the relatively coarse-grained fraction of fine-grained sediments, whereas Rb is associated with the fine-grained fraction; (f) D50 represents the maximum diameter of 50% of the sediment grain size. These plots are correlated to reconstruction of NAO (h) from a lake in Greenland (Olsen et al., 2012) and reconstructed SST (C°) from alkenones (j) in the same core (Jalali et al., 2016). Blue bands correspond to CR0-6 chronology, dotted black lines highlight the main "*wet*" events that may be observed in sediment records in the Late Holocene (Table 3).

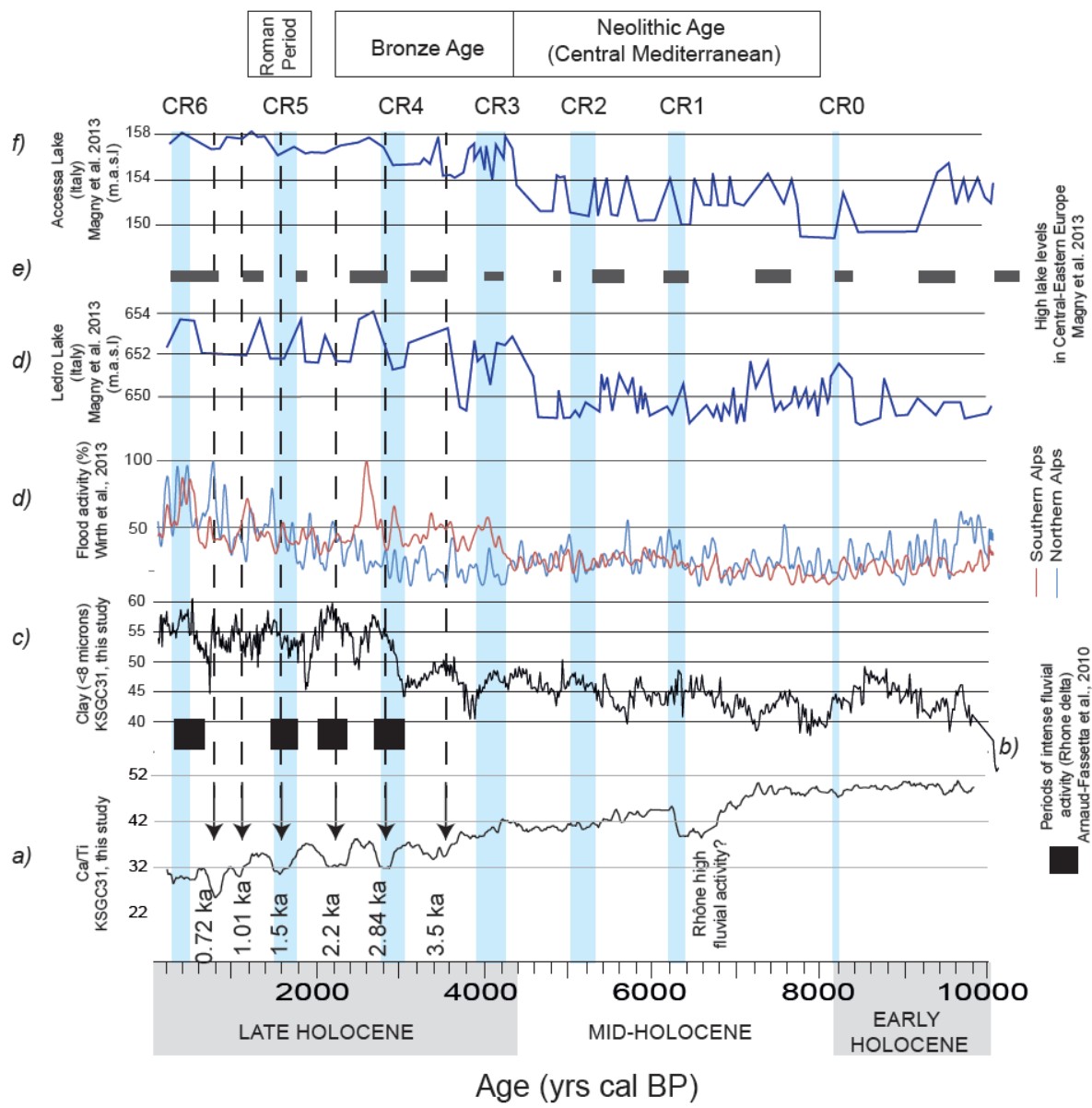

**Figure 5:** Ca/Ti ratio (a) and percentage of fine-grained (<8µm) sediment (c) compared to: b) Periods of intense fluvial activity based on hydromorphological and paleohydrological changes in the Rhone delta (Arnaud-Fassetta et al., 2010); d) Holocene flood frequency (%) in Southern and Northern Alps estimated on the basis on lake flood records by Wirth et al. (2013); e); f); g) lake level fluctuations in Central and Eastern Europe during the Holocene (Magny et al., 2013). Blue bands correspond to CR0-6 chronology, dotted black lines highlight the main "*wet*" events that may be observed in sediment records in the Late Holocene (Table 3).