# Peer review of "Holocene hydrological changes of the Rhone River (NW Mediterranean) as recorded in the marine mud belt"

_Climate of the Past, 2016_

## Referee Comment (RC1) · Anonymous Referee #1 · 22 Mar 2016

This paper is an excellent example of integrated high resolution sedimentological and geochemical study feeding the data bank with valuable paleoclimatic / paleohydrological proxies established on a precise and reliable time scale fixed on highly accurate calibrated C-14 ages. The sediment column and the data obtained are of exceptional quality and the interpretation is well detailed and thoroughly presented. I recommend publication without significant corrections. Few comment however: - When citing many successive refs (frequent in this paper) you should avoid redundant authors citations (cite for example the princeps, or the most significant) or develop in order to distinguish their respective contributions. Many parts of this paper heavily suffer from this default. The refs list is therefore far too long. Counter examples : only one ref for the 8.2 ka cold

event and this is not the princeps one. No citation of the African humid period in the introduction ( lines 40 ...) while the discussion obviously evokes this major Holocene regional climate trend.

In section 4, present first the seismic profile ( figure 2) and then the age model ( figure 3)... lines 295-307 should be displaced at line 280. Note that the age model is presented twice at few line differences: end of methods beginning of results ...

details: lines 235-236: precise the concept: relatively coarse grain fraction of fine grained sediment... ? relativist but not such obvious ! Avoid S.R for sedimentation rate : you use it once: useless!

section 5 should be named: interpretation and discussion.

---

## Short Comment (SC1) · 13 Apr 2016

This is an important long Holocene dataset which deserves publishing.

It would be good if the authors could add some explanations to Figure 4 about the significance of the various coloured zones (blue, white, light grey, dark grey). I have not found this in the legend.

Another observation concerns the Medieval Climate Anomaly (MCA). The Ca/Ti curve shows a major negative low peak 1200-600 yrs BP (800-1400 AD). Low Ca/Ti values are interpreted as high precipitation, i.e. a wet phase. The MCA therefore was wet, which matches with the authors' statement in line 449 of the manuscript. In figure 4,

however, the Little Ice Age (LIA) is shown at 1300 AD. This is not the typical timing of the LIA which has its key phase 1400-1850 AD. Looking at this period, the Ca/Ti is high, therefore climate dry. The LIA therefore was dry. This finding could be added to the discussion part of the paper. I would also change the figure so that the LIA is positioned in the right place.

―――――――――――――――――――――――

---

## Short Comment (SC2) · 13 Apr 2016

I forgot to mention one observation with regards to the Medieval Climate Anomaly (MCA). As mentioned, the MCA 1100-600 yrs BP is marked by a wet phase (low Ca/Ti). Interestingly, the grain sizes (as measured by D50) has a high peak during this period, i.e. grain sizes has gone up. It therefore seemt to me that this wet phase is associated with coarser grains. This is the exact opposite of which has been stated by the authors in line 435 in the manuscript. How does this fit together? In comparison, the (dry) Little Ice Age (sensu stricto) 600-150 yrs BP has relatively fine-grained strata. It seems to me that at least during MCA times, more rain means more sediment has been transported which was also coarser than usual. When LIA climate was drier, the coarse material

could not be transported so easily and only the mud made it into the offshore core location. What do you think about this scenario?

---

## Short Comment (SC3) · 25 Apr 2016

Dear colleague, thank you for your comments. Concerning the grain size anomaly, you have raised a good point. Yes, the MCA mostly falls into relatively high D50 values, thus higher grain size. Your proposed scenario (more rain= more sediment input, coarser fraction transported over longer distance/ less rain= coarse sediment retained at the fluvial source) makes perfectly sense. The line 435 statement will be revised accordingly.

---

## Referee Comment (RC2) · Anonymous Referee #2 · 27 Apr 2016

This manuscript presents new data from a marine sediment core from the Gulf of Lions (W Mediterranean Sea) that allows to evaluate the Holocene evolution of river discharges with an exceptional high resolution for a marine record. The data set covers the Holocene period with an accurate chronology based on 18 AMS 14C ages. The presented data correspond to XRF-scanner element ratios combined with grain-size and element (C and N) measurements. They are interpreted in terms of changes in the intensity of river runoff, concretely from the Rhone river (south France). The manuscript discusses the results in the context of other paleoclimatic records, providing a very wide and well documented perspective of the currently published literature in the subject. I consider that the quality of the data and the well documented discussion

are good to be published in Climate of the Past, but I think that the discussion needs some modification before its publication in Climate of the Past. There are few minor aspects that I mention bellow but in general there are two important elements that I consider the discussion would need to include before its final publication:

One of the main conclusions of the manuscript is that from 11 to 4 ka the terrigenous imput/humidity progressively increased reaching maximum values at 7ka (see bellow my specific comments on this particular event). Looking to the presented data, this evolution is particularly clear in the Ca/Ti and CaCO3 data. Other proxies such as K/Ti present a rather stable base line with several interesting low frequency oscillation but, overall, it shows higher values (more humid, according to author's interpretation) during early Holocene. I think that authors need to argue better why they believe that low "detritic" carbonate is a good proxy for more humid conditions. The Ti/Ca interpretation as a terrestrial proxy is clear in areas where the Ca signal is dominated by primary productivity but here the authors interpret that Ca has mostly a detritic origin. This point is also relevant for the grainsize measurements (see my specific comments bellow). It is particular relevant to support well the interpretation of the humidity proxies in this case, since Holocene humidity evolution in this region is a matter of a current debate. The interpreted evolution in this manuscript is contrasting with the overall view of Holocene humidity evolution in the Mediterranean region, with more humid conditions during the early than the late Holocene (see as a reviews: Sadori et al., 2011; Jalut et al., 2009). This is a view mostly based in pollen sequences including some from the South of France. Nerveless, this overall Holocene humidity patter is currently under discussion since some lake level records (some shown in Fig 5) indicate a rather humid late Holocene (Magny et al., 2013). This is a still open discussion and this different proxy/regional trends in humidity along the Holocene should be discussed also in the manuscript and better argued the interpretation of the chosen proxy.

The second aspect that should be consider in the discussion is the about the general focus around the global events defined by Wanner et al. (2011). Some of these events

occurred during dry, other during wet or during transient conditions when they are compared with the new presented data. This absence of consistency in patters also occurs when these "global" events are compared with the SST record, they do not always match with cold conditions in the Gulf of Lions. This is not surprising if we consider the enormous geographical heterogeneity in the time distribution of these events according to Figure 2a in Wanner et al (2011). The definition of these events is an interesting and useful exercise to try to determine the most robust patterns at global scale but they are not good templates to explain regional variability. I say this because focussing the discussion so much in these "global" events the manuscript often ignores the actual and interesting signal of the records presented in the manuscript. This is a real problem in figures 4 and 5 where the blue bars concentrate the attention and they do not actually highlight the relevant patterns in the presented records, not even in fig 5 when the new records are compared with others, these blue bars become a noise that difficult the comparison of the actual data presented in the manuscript. That becomes even more complex with other black lines drawn in figures 4 and 5 which do not necessarily coincide with the blue bars. Figure caption does not help with the bars/line reading since do not describe what they are. After the manuscript reading black lines seems to highlight the wet episodes interpreted from the Ca/Ti and/or K/Ti records. These should be the events to be well highlighted in the figures. It is not so relevant to locate the CR from Wanner et al. (2011), particularly when any coherent pattern is presented in the region, these were temperature events no precipitation events. It is not a problem that there is not a coherent pattern between these "global" events and the studied records. It is far more relevant to better compare the found wet events in the new records with the other regional humidity records presented in Fig 5. One aspect that is not consider in the manuscript and could be explored is to analyse the frequency of the humidity changes, particularly with the K/Ti record, the good chronology and high resolution would allow to do a solid analyse, similar analyses in other W Mediterranean records have proposed an middle Holocene change in the frequency of humidity events (Fletcher et al. 2012). That could be further explored with the presented records since the presented records

have high resolution and solid chronology to test this hypothesis. In this line, figures also highlight very much the three "formal" Holocene periods. I agree with authors that it is important to show them in the figure but I would probably not mark them so much with different grey bars over the graphs. Again, the major changes in the record evolution are not happening within these sub-stages limits and they disturb the attention looking for these limits in the records.

Specific comments:

150: "The Mediterranean hydrology is controlled by the seasonality of precipitation as well as the catchment geology, vegetation type and geomorphology of the region."

This sentence is not entirely true, catchment geology, vegetation and geomorphology will control the characteristics of the sedimentary charge of the rivers and thus transport and sedimentation of terrigenous sediments in the ocean but they won't control the hydrology of the Mediterranean.

182: ibid. What does it means?

Section 3: According to the methodological description samples were not decarbonated for grain-size analysis, according to figure 4, the Ca signal is interpreted to reflect mostly detritic carbonate. It would be useful in this section to do a brief description about the richness and characteristic of the biogenic fraction in these sediments. According with the manuscript assumptions they should be negligible or very low in relation with the detritic carbonate, but that should be described in this section.

275: Here the CR1 and CR6 are introduced, but they need to be defined "Cold Relapses??" and indicate the source (Wanner et al 2011??). They are actually defined later on in line 284 but they need to be defined the first time they are used and referenced properly.

380: Note that the full reference of Jalali et al is not included in the reference list.

384: It is not clearly described why the Ca/Ti is used as a proxy for terrigenous input.

It is indeed the given interpretation in several oceanographic settings but in this case is not so clear since according to figure 4v the carbonate in this core is interpreted to be detritic carbonate. It should be argued somewhere why it is believed that Ti/Ca can be used in this case as a proxy of terrestrial input since both elements are supposed to have the same terrestrial source.

388: Here the discussion focuses very much the attention in an event at 7-6.4 kyr, which is tentatively correlated to the SHP (Saharan Humid Period). This term was new to me and I have searched in the cited literature to check where it was previously defined and I could not find it. I guess that authors refer to the African Humid Period (AHP) but it was not an event from 7-6.4 ky as appears in the text and figures, it was a far longer period (11.7-5 ky). One of the used references (Castañeda et al., 2016) describes a peak maximum for this AHP but its chronology and duration is also different (9-7 ka) to that of this manuscript event. I would suggest to remove the reference to this Sahara Humid Period which does not exist in the literature with this name and chronology. The discussion focuses some attention in this event which only appears as a relevant event in the Ca/Ti record. K/Ti record does not show any relevant structure, neither Zr/Rn, not D50 nor the SST. The text argues that this was a period of finer material (D50) but it is not particularly finer in the record when it is compared with other previous or later events. From the chronological point of view does not make much sense to connect this event with the AHP which was far longer in time extend, but it is also an unclear the proposed connection from the mechanistic point of view. It is very true that, at orbital time scales, changes in the seasonal/latitude energy valance have been interpreted to stimulate both African monsoon and precipitation over the Mediterranean at different seasons. But this is not a direct cause-effect between this two rain regimes, it is a response to the orbital forced changes in insolation (Bosmans et al., 2015 and Kutzbach et al., 2013). Orbital driven changes should not be clamed to explain centennial scale events.

417: Why focussing the attention in what is not in the record? Better concentrate in

what is actually present in the record.

448: MCA in the Mediterranean region has been described as an overall dry period, and this sentence here gives the impression that in this record is wet. Looking carefully to the record it appears that the wet conditions refer to the short episodes before, after and one in the middle, but overall records indicate rather dry conditions. This sentence should probably be rephrased to avoid confusion giving the wrong impression that the MCA here was a wet episode.

Used references: Bosmans, J. H. C., Drijfhout, S. S., Tuenter, E., Hilgen, F. J., Lourens, L. J., & Rohling, E. J. (2015). Precession and obliquity forcing of the freshwater budget over the Mediterranean. Quaternary Science Reviews, 123, 16–30. doi:10.1016/j.quascirev.2015.06.008

Castañeda, I. S., Schouten, S., Pätzold, J., Lucassen, F., Kasemann, S., Kuhlmann, H., & Schefuß, E. (2016). Hydroclimate variability in the Nile River Basin during the past 28,000 years. Earth and Planetary Science Letters, 438, 47–56. doi:10.1016/j.epsl.2015.12.014

Fletcher, W. J., Debret, M., & Sanchez Goñi, M. F. (2012). Mid-Holocene emergence of a low-frequency millennial oscillation in western Mediterranean climate: Implications for past dynamics of the North Atlantic atmospheric westerlies. The Holocene. doi:10.1177/0959683612460783

Jalut, G., Dedoubat, J. J., Fontugne, M., & Otto, T. (2009). Holocene circum-Mediterranean vegetation changes: Climate forcing and human impact. Quaternary International, 200(1-2), 4.

Kutzbach, J. E., Chen, G., Cheng, H., Edwards, R. L., & Liu, Z. (2013). Potential role of winter rainfall in explaining increased moisture in the Mediterranean and Middle East during periods of maximum orbitally-forced insolation seasonality. Climate Dynamics, 42(3), 1079–1095. doi:10.1007/s00382-013-1692-1

[Figure]

Magny, M., Combourieu-Nebout, N., de Beaulieu, J. L., Bout-Roumazeilles, V., Colombaroli, D., Desprat, S., ... Wirth, S. (2013). North-south palaeohydrological contrasts in the central Mediterranean during the Holocene: tentative synthesis and working hypotheses. Climate of the Past, 9(5), 2043–2071. doi:10.5194/cp-9-2043-2013

Sadori, L., Jahns, S., & Peyron, O. (2011). Mid-Holocene vegetation history of the central Mediterranean. The Holocene, 21(1), 117–129. doi:10.1177/0959683610377530

Wanner, H., Solomina, O., Grosjean, M., Ritz, S. P., & Jetel, M. (2011). Structure and origin of Holocene cold events. Quaternary Science Reviews, 30(21'Äì22), 3109–3123. doi:10.1016/j.quascirev.2011.07.010
* * *

---

## Author Comment (AC1) · 2 Jun 2016

REPLAY to:

**Anonymous Referee #1**

This paper is an excellent example of integrated high resolution sedimentological and geochemical study feeding the data bank with valuable paleoclimatic / paleohydrological proxies established on a precise and reliable time scale fixed on highly accurate calibrated C-14 ages. The sediment column and the data obtained are of exceptional quality and the interpretation is well detailed and thoroughly presented. I recommend publication without significant corrections. Few comment however: -
 When citing many successive refs (frequent in this paper) you should avoid redundant authors citations (cite for example the princeps, or the most significant) or develop in order to distinguish their respective contributions. Many parts of this paper heavily suffer from this default.
The refs list is therefore far too long. Counter examples : only one ref for the 8.2 ka cold event and this is not the princeps one.

The reference list has been reduced, the less significant references have been removed.

No citation of the African humid period in the introduction ( lines 40 ...) while the discussion obviously evokes this major Holocene regional climate trend.

The reference to AHP was removed, according the suggestion given by Referee 2

In section 4, present first the seismic profile (figure 2) and then the age model ( figure 3)...
lines 295-307 should be displaced at line 280. Note that the age model is presented twice at few line differences: end of methods beginning of results ...
This is corrected in the revised version
details:  lines  235-236: precise the concept: relatively coarse grain fraction of fine grained sediment... ? relativist but not such obvious !
Corrected in the revised version

Avoid S.R for sedimentation rate : you use it once: useless!
Actually, we have used the SR for "sedimention rate" all along the manuscript. I take the freedom to not change it.

section 5 should be named: interpretation and discussion.
Corrected

---

## Author Comment (AC2) · 2 Jun 2016

This manuscript presents new data from a marine sediment core from the Gulf of Lions (W Mediterranean Sea) that allows to evaluate the Holocene evolution of river discharges with an exceptional high resolution for a marine record. The data set covers the Holocene period with an accurate chronology based on 18 AMS 14C ages. The presented data correspond to XRF-scanner element ratios combined with grainsize and element (C and N) measurements. They are interpreted in terms of changes in the intensity of river runoff, concretely from the Rhone river (south France). The manuscript discusses the results in the context of other paleoclimatic records, providingaverywideandwelldocumentedperspectiveofthecurrentlypublishedliteraturein the subject. I consider that the quality of the data and the well documented discussion are good to be published in Climate of the Past, but I think that the discussion needs some modification before its publication in Climate of the Past. There are few minor aspects that I mention bellow but in general there are two important elements that I consider the discussion would need to include before its final publication: One of the main conclusions of the manuscript is that from 11 to 4 ka the terrigenous imput/humidity progressively increased reaching maximum values at 7ka (see bellow my specific comments on this particular event). Looking to the presented data, this evolution is particularly clear in the Ca/Ti and CaCO3 data. Other proxies such as K/Ti present a rather stable base line with several interesting low frequency oscillation but, overall, it shows higher values (more humid, according to author's interpretation) during early Holocene. I think that authors need to argue better why they believe that low "detritic" carbonate is a good proxy for more humid conditions. The Ti/Ca interpretation as a terrestrial proxy is clear in areas where the Ca signal is dominated by primary productivity but here the authors interpret that Ca has mostly a detritic origin. This point is also relevant for the grainsize measurements (see my specific comments bellow).

The two points resumed in this introduction have been revised, according the specific comments given below.

It is particular relevant to support well the interpretation of thehumidity proxies in this case, since Holocene humidity evolution in this region is a matter of a current debate. The interpreted evolution in this manuscript is contrasting with the overall view of Holocene humidity evolution in the Mediterranean region, with more humid conditions during the early than the late Holocene (see as a reviews: Sadori et al., 2011; Jalut et al., 2009). This is a view mostly based in pollen sequences including some from the South of France. Nerveless, this overall Holocene humidity patter is currently under discussion since some lake level records (some shownin Fig 5) indicate a rather humid late Holocene (Magny et al., 2013). This is a still open discussion and this different proxy/regional trends in humidity along the Holocene should be discussed also in the manuscript and better argued the interpretation of the chosen proxy.

The second aspect that should be consider in the discussion is the about the general focus around the global events defined by Wanner et al. (2011). Some of these events occurred during dry,other during wet or during transient conditions when they are compared with the new presented data. This absence of consistency in patters also occurs when these "global" events are compared with the SST record, they do not always match with cold conditions in the Gulf of Lions.

This is not surprising if we consider the enormous geographical heterogeneity in the time distribution of these events according to Figure 2a in Wanner et al (2011). The definition of these events is an

interesting and useful exercise to try to determine the most robust patterns at global scale but they are not good templates to explain regional variability. I say this because focussing the discussion so much in these "global" events the manuscript often ignores the actual and interesting signal of the records presented in the manuscript.

*The discussion about the "global" meaning of cold relapses have been clarified. It is well known that the local response of cold events can be different according to regions. I have integrated a supplementary comment in the discussion (section 5) "Numerous forcing factors (i.e. sea-level, ice cap extent, forest cover, volcanic activity, etc) may account for the climate variability in the Holocene. Statistical analysis of proxy time series in both northern and southern hemisphere (Wanner et al., 2011) have demonstrated that multidecadal to multicentury cold relapses (CRs) interrupted periods of relative stable climate conditions. They are demonstrated to exist at least in the North Atlantic (Bond, 1997) and surrounding land areas. Thus, there is a general agreement about the different local expressions and timing offset of these rapid climate changes according to geographical position or geomorphological setting. In a way, these events cannot be considered as really global, but they nonetheless represent significant milestones in the Holocene climate history. In this paper, we use the correlation with CRs in order to highlight possible differences in features and chronology of rapid events between Atlantic and western Mediterranean during Early, Middle and Late Holocene.*

This is a real problem in figures 4 and 5 where the blue bars concentrate the attention and they do not actually highlight the relevant patterns in the presented records, not even in fig 5 when the new records are compared with others, these blue bars become a noise that difficult the comparison of the actual data presented in the manuscript. That becomes even more complex with other black lines drawn in figures 4 and 5 which do not necessarily coincide with the blue bars. Figure caption does not help with the bars/line reading since do not describe what they are. After the manuscript reading black lines seems to highlight the wet episodes interpreted from the Ca/Ti and/or K/Ti records. These should be the events to be well highlighted in the figures.

*Figures have been simplified*

It is not so relevant to locate the CR from Wanner et al. (2011), particularly when any coherent pattern is presented in the region, these were temperature events no precipitation events. It is not a problem that there is not a coherent pattern between these "global" events and the studied records. It is far more relevant to better compare the found wet events in the new records with the other regional humidity records presented in Fig 5.

*We have simplified the figure, but kept the CRs blue bars. We believe that showing the chronology of the CRs in the Atlantic still is important, for comparison (see comment above)*

One aspect that is not consider in the manuscript and could be explored is to analyze the frequency of the humidity changes, particularly with the K/Ti record, the good chronology and high resolution would allow to do a solid analyse, similar analyses in other W Mediterranean records have proposed an middle Holocene change in the frequency of humidity events (Fletcher et al. 2012). That could be further explored with the presented records since the presented records have high resolution and solid chronology to test this hypothesis.

*This suggestion is surely interesting, but we have chosen to not use the spectral analysis approach. The main raison is explained in the last part of the discussion (from line 457 in the modified text): the degree of chemical weathering (given by the Ki/Ti) mainly depends on the vegetation cover, certainly, but the anthropic impact during the last 4000 years can be significant in the Rhone valley and delta plain. Thus, a "pure" climate signal is difficult to obtain at this site. We need at least to extract the "anthropogenic factor", and we are not able to do it at the moment.*

In this line, figures also highlight very much the three "formal" Holocene periods. I agree with authors that it is important to show them in the figure but I would probably not mark them so much with different grey bars over the graphs. Again, the major changes in the record evolution are not happening within these sub-stages limits and they disturb the attention looking for these limits in the records.

Figures have been simplified and supplementary explanations have been added

Specific comments: 150: "The Mediterranean hydrology is controlled by the seasonality of precipitation as well as the catchment geology, vegetation type and geomorphology of the region." This sentence is not entirely true, catchment geology, vegetation and geomorphology willcontrolthecharacteristicsofthesedimentarychargeoftheriversandthustransport and sedimentation of terrigenous sediments in the ocean but they won't control the hydrology of the Mediterranean.

The sentence has been changed *"The hydrological budget in the Mediterranean borderlands depends on the seasonality of precipitation as well as the catchment geology, vegetation type and geomorphology of the region."*

182: ibid. What does it means?

I tried to make it clearer: *"The disconnection between the river and the canyon head is dated at 19 ka cal BP in response to rapid sea-level rise (ibid.). The landwards retreat path of the estuary mouth on the shelf has be tracked through the mapping and dating of paleo-delta lobes"*

Section 3: According to the methodological description samples were not decarbonated for grain-size analysis, according to figure 4, the Ca signal is interpreted to reflect mostly detrital carbonate. It would be useful in this section to do a brief description about the richness and characteristic of the biogenic fraction in these sediments. According with the manuscript assumptions they should be negligible or very low in relation with the detrital carbonate, but that should be described in this section.

The significance of "detritic carbonate" has been clarified: *"… it is worthwhile reminding that calcite of detritic origin, generated by erosion of calcareous massifs in the catchment area, represents an important component of the fluvial Rhône waters sediment load. This type of calcite is transported into the sea but it is mainly accumulated in the sand fraction, trapped in the proximal deltaic sediments. In the mud belt where deposits are mostly pelitic, the detritic calcite quickly decreases seaward of the river mouth, with only a very small fraction being preserved in the clay fraction (Chamley, 1971). On the other hand, calcite of biogenic marine origin (bioclasts) is usually abundant. Benthic (rare planktonic) foraminifera, ostracods, fragmented mollusk shells and debris from bryozoan and echinoids can be observed under the binocular microscope. Thus, the Ca content in the core KSGC31 is considered as related to biogenic marine productivity"*

275: Here the CR1 and CR6 are introduced, but they need to be defined "Cold Relapses??" and indicate the source (Wanner et al 2011??). They are actually defined later on in line 284 but they need to be defined the first time they are used and referenced properly.

It is done in the text

380: Note that the full reference of Jalali et al is not included in the reference list.

The full reference has been added

384: It is not clearly described why the Ca/Ti is used as a proxy for terrigenous input.

It is indeed the given interpretation in several oceanographic settings but in this case is not so clear since according to figure 4v the carbonate in this core is interpreted to be detritic carbonate. It should be argued somewhere why it is believed that Ti/Ca can be used in this case as a proxy of terrestrial input since both elements are supposed to have the same terrestrial source.

See the comment to Section 3 (above), it clarifies the use of Ca/Ti proxy

388: Here the discussion focuses very much the attention in an event at 7-6.4 kyr, which is tentatively correlated to the SHP (Saharan Humid Period). This term was new to me and I have searched in the cited literature to check where it was previously defined and I could not find it. I guess that authors refer to the African Humid Period (AHP) but it was not an event from 7-6.4 ky as appears in the text and figures, it was a far longer period (11.7-5 ky). One of the used references (Castañeda et al., 2016) describes a peak maximum for this AHP but its chronology and duration is also different (9-7 ka) to that of this manuscript event. I would suggest to remove the reference to this Sahara Humid Period which does not exist in the literature with this name and chronology. The discussion focuses some attention in this event which only appears as a relevant event in the Ca/Ti record. K/Ti record does not show any relevant structure, neither Zr/Rn, not D50 nor the SST. The text argues that this was a period of finer material (D50 )but it is not particularly finer in the record when it is compared with other previous or later events. From the chronological point of view does not make much sense to connect this event with the AHP which was far longer in time extend, but it is also an unclear the proposed connection from the mechanistic point of view. It is very true that, at orbital time scales, changes in the seasonal/latitude energy valance have been interpreted to stimulate both African monsoon and precipitation over the Mediterranean at different seasons. But this is not a direct cause-effect between this two rain regimes, it is a response to the orbital forced changes in insolation (Bosmans et al., 2015 and Kutzbach et al., 2013). Orbital driven changes should not be claimed to explain centennial scale events.

The reference to African (Saharan) Humid Period has been removed. Yes, it is too speculative, the timing is too offset. We agree that it was not a good idea!

417: Why focusing the attention in what is not in the record? Better concentrate in what is actually present in the record.

I have rephrased: *"According to our data, during CR1 and CR2, chemical weathering was weak as suggested by high K/Ti values (Figure 4d), mean SR was generally low (<1 mm/yr on average, Table 2) but there was no significant change in terrigenous inputs (Ca/Ti ratio, Figure 4c) . The C:Na ratios indicate better preservation of nitrogen organic compounds preferentially adsorbed in the clay fraction (Figure 4a). The reduction of the vegetation cover in the river catchment, combined with lower (1-1.5°C) temperatures in the European Alps (Haas et al., 1998), may explain the low chemical degradation state of the illite minerals. A drop in sea surface temperature is also recorded by alkenones in the core as illustrated in Jalali et al. (2016), confirming the impact of the cold relapses in the Mediterranean area in the Mid- Holocene (Figure 4i)."*

448: MCA in the Mediterranean region has been described as an overall dry period, and this sentence here gives the impression that in this record is wet. Looking carefully to the record it appears that the wet conditions refer to the short episodes before, after and one in the middle, but overall records indicate rather dry conditions. This sentence should probably be rephrased to avoid confusion giving the wrong impression that the MCA here was a wet episode.

It has been rephrased " *In the KSGC31, these events (~2840 a, ~ 1500 and ~720 a cal BP, Figure 4) barely coincide with the cold events in the North Atlantic but are concomitant with periods of*

*increasing flood frequency in the Northern Alps as reconstructed by Wirth et al. (2013) (Figure 5d) and punctuated overall warm (and dry) periods such as the MCA at ~ 3,500; ~ 2,200 and ~1,000 and ~0,72 a cal BP (Figure 4). This pattern suggest different causes for enhanced precipitations in the late Holocene"*

---

## Author Comment (AC3) · 2 Jun 2016

**Short Comment 1**

This is an important long Holocene dataset which deserves publishing.
It would be good if the authors could add some explanations to Figure 4 about the significance of the various coloured zones (blue, white, light grey, dark grey). I have not found this in the legend.

I have simplified and added some explications in the figure 4

Another observation concerns the Medieval Climate Anomaly (MCA). The Ca/Ti curve shows a major negative low peak 1200-600 yrs BP (800-1400 AD). Low Ca/Ti values are interpreted as high precipitation, i.e. a wet phase. The MCA therefore was wet, which matches with the authors' statement in line 449 of the manuscript.

This point was slightly modified according the suggestion given by Reviewer 2. The MCA is not "humid", but overall dry and punctuated by short wet periods.

In figure 4, however, the Little Ice Age (LIA) is shown at 1300 AD. This is not the typical timing of the LIA which has its key phase 1400-1850 AD. Looking at this period, the Ca/Ti is high, therefore climate dry. The LIA therefore was dry. This finding could be added to the discussion part of the paper. I would also change the figure so that the LIA is positioned in the right place.

LIA has been replaced. Revising the LIA timing and data I have reached a different conclusion: *"[…] Episodes of enhanced terrigenous inputs (during floods, for instance) are detected by low Ca/Ti ratios that also coincide with low Zr/Rb and low D50 values, indicating general smaller-size terrigenous grains as also suggested by high clay content (Figure 5c). Indeed, after the stabilization of sea-level, only the finest sediment fraction (clay) transported by the river plume reaches the mud belt at the core site. An exception to this pattern is observed for the LIA when quite high Ca/Ti would suggest relatively "dry" conditions (Figure 4c,f). The qualitative observation under the binocular microscope reveals the presence (only in this specific interval) of abundant bryozoans and Elphidium crispum (coastal benthic foraminifer) tests together with rare grains of quartz, which compose the coarser (>63 µm) fraction. The biogenic debris can explain the high Ca content and presence of quartz grain, the peak of Zr/Rb (Figure 4e). The accumulation of this material is maybe due to concomitant occurrence of river floods (Figure 5d) and storms, which might have remobilized coarse material from coastal setting (Bourrin et al., 2015)"*

**Short Comment 2**

I forgot to mention one observation with regards to the Medieval Climate Anomaly (MCA). As mentioned, the MCA 1100-600 yrs BP is marked by a wet phase (low Ca/Ti). Interestingly, the grain sizes (as measured by D50) has a high peak during this period, i.e. grain sizes has gone up. It therefore seem to me that this wet phase is associated with coarser grains. This is the exact opposite of which has been stated by the authors in line 435 in the manuscript. How does this fit together?
In comparison, the (dry) Little Ice Age (sensu stricto) 600-150 yrs BP has relatively fine-grained strata. It seems to me that at least during MCA times, more rain means more sediment has been transported which was also coarser than usual. When LIA climate was drier, the coarse material

could not be transported so easily and only the mud made it into the offshore core location. What do you think about this scenario?
See previous comment.